# Visfatin-Induced Inhibition of miR-1264 Facilitates PDGF-C Synthesis in Chondrosarcoma Cells and Enhances Endothelial Progenitor Cell Angiogenesis

**DOI:** 10.3390/cells11213470

**Published:** 2022-11-02

**Authors:** Chang-Yu Song, Sunny Li-Yun Chang, Chih-Yang Lin, Chun-Hao Tsai, Shang-Yu Yang, Yi-Chin Fong, Yu-Wen Huang, Shih-Wei Wang, Wei-Cheng Chen, Chih-Hsin Tang

**Affiliations:** 1Graduate Institute of Biomedical Science, China Medical University, Taichung 404333, Taiwan; 2Translational Medicine Center, Shin-Kong Wu Ho-Su Memorial Hospital, Taipei City 111045, Taiwan; 3Department of Sports Medicine, College of Health Care, China Medical University, Taichung 404333, Taiwan; 4Department of Orthopedic Surgery, China Medical University Hospital, Taichung 404333, Taiwan; 5Department of Healthcare Administration, College of Medical and Health Science, Asia University, Taichung 400354, Taiwan; 6Department of Orthopaedic Surgery, China Medical University Beigang Hospital, Yunlin 651012, Taiwan; 7Department of Medicine, MacKay Medical College, New Taipei City 25245, Taiwan; 8Institute of Biomedical Sciences, Mackay Medical College, Taipei 25245, Taiwan; 9School of Pharmacy, College of Pharmacy, Kaohsiung 807378, Taiwan; 10Division of Sports Medicine & Surgery, Department of Orthopedic Surgery, MacKay Memorial Hospital, Taipei 25245, Taiwan; 11Department of Pharmacology, School of Medicine, China Medical University, Taichung 404333, Taiwan; 12Chinese Medicine Research Center, China Medical University, Taichung 404333, Taiwan; 13Department of Biotechnology, College of Health Science, Asia University, Taichung 400354, Taiwan

**Keywords:** chondrosarcoma, angiogenesis, endothelial progenitor cell, visfatin, tumor angiogenesis, platelet-derived growth factor, PDGF-C

## Abstract

New treatments for chondrosarcoma are extremely important. Chondrosarcoma is a primary malignant bone tumor with a very unfavorable prognosis. High-grade chondrosarcoma has a high potential to metastasize to any organ in the body. Platelet-derived growth factor (PDGF) is a potent angiogenic factor that promotes tumor angiogenesis and metastasis. The adipocytokine visfatin promotes metastatic potential of chondrosarcoma; however, the role of visfatin in angiogenesis in human chondrosarcoma is unclear. We report that the levels of PDGF-C expression were positively correlated with tumor stages, significantly higher than the levels of expression in normal cartilage. Visfatin increased PDGF-C expression and endothelial progenitor cell (EPC) angiogenesis through the PI3K/Akt/mTOR signaling pathway, and dose-dependently down-regulated the synthesis of miR-1264, which targets the 3′-UTR of PDGF-C. Additionally, we discovered inhibition of visfatin or PDGF-C in chondrosarcoma tumors significantly reduced tumor angiogenesis and size. Our results indicate that visfatin inhibits miR-1264 production through the PI3K/Akt/mTOR signaling cascade, and thereby promotes PDGF-C expression and chondrosarcoma angiogenesis. Visfatin may be worth targeting in the treatment of chondrosarcoma angiogenesis.

## 1. Introduction

Chondrosarcoma is well known for responding poorly to chemotherapy and radiotherapy [1,2,3]. Histopathologic grades of chondrosarcoma categorize the tumors by aggressiveness and disease prognosis, according to hematoxylin and eosin (H&E) staining results that identify chondrosarcomas as grade I (low-grade), grade II (intermediate-grade), or grade III (high-grade), according to increasing cellularity, matrix proteins, and mitotic activity [4]. New biological markers to distinguish grades of chondrosarcoma are essential [5]. The proinflammatory adipocytokine visfatin is the main rate-limiting enzyme in the synthesis of cellular nicotinamide adenine dinucleotide (NAD^+^) and an important regulator of mammalian cell proliferation, angiogenesis, and apoptosis [6]. Importantly, researchers have reported finding increased levels of visfatin in chondrosarcoma cartilage [7,8]; visfatin expression increases with higher grades of chondrosarcoma, with the highest levels of visfatin found in grade III (poorly differentiated) disease. Visfatin promotes angiogenesis activity in endothelial cells and cancer cells and also chondrosarcoma metastasis [9]. We therefore sought to determine the relationships between visfatin and chondrosarcoma angiogenesis in cellular and preclinical experiments.

Platelet-derived growth factor (PDGF) and vascular endothelial growth factor (VEGF) are key factors in angiogenesis [10,11]. PDGF promotes angiogenesis by enhancing VEGF and PDGF signaling and *VEGF* gene expression and activating PDGF receptors (PDGFRs), or by upregulating stromal *VEGF* [12]. In particular, PDGF-C is a potent mediator of cell migration, growth, survival, transformation, apoptosis, tissue development, and wound healing; stimulates Ewing’s sarcoma proliferation; and induces paracrine signaling events in whole tumor lysates [13,14]. Moreover, a significant positive correlation exists between PDGFR-α and chondrosarcoma tumor aggressiveness [15]. We therefore sought to determine whether visfatin affects PDGF-C-dependent angiogenesis in chondrosarcoma and to clarify the underlying mechanisms.

MicroRNAs (miRNAs), small non-coding RNAs capable of post-transcriptional regulation of gene synthesis [16,17], affect cancer biology and pathology [18]. MiRNAs have been used as biomarkers in tumor-related diagnoses and prognoses [19], and their progression and treatment, with evidence showing abnormal miRNA expression in a variety of different cancers, including chondrosarcoma, where the abnormal upregulation or downregulation of several miRNAs promote its progression [20,21,22]. MiR-1264 has been reported to be dysregulated in colorectal cancer, laryngeal cancer, and neuroglioma [23,24]. Recently, a report also indicated miR-1264 regulates progression of breast cancer [25]. However, no existing evidence has confirmed any regulatory activity by the PDGF-C–miRNA axis during angiogenesis in chondrosarcoma. Our study demonstrates that visfatin enhances PDGF-C production and facilitates EPC angiogenesis in human chondrosarcoma cells by inhibiting miR-1264 production in the PI3K/Akt/mTOR signaling pathway. Inhibiting visfatin downregulated in vivo PDGF-C-dependent angiogenesis. We suggest that visfatin is worth targeting as a therapeutic strategy for chondrosarcoma.

## 2. Materials and Methods

### 2.1. Cell Lines

The human chondrosarcoma cell line SW1353 was supplied by the American Type Cell Culture Collection; the JJ012 cell line was gifted by Dr. Sean P. Scully (University of Miami School of Medicine, Miami, FL, USA). JJ012 cells stably expressing visfatin (JJ012/visfatin cells) and JJ012/visfatin cells with knockdown PDGF-C (JJ012/visfatin/shPDGF-C cells) were established in complementary DNA (cDNA) 24 h after transfection; stable transfectants were selected in G418 (Geneticin) at a concentration of 200 μg/mL. Cells were cultured in DMEM/α-MEM supplemented with 10% FBS and 100 units/mL penicillin/streptomycin at 37 °C with 5% CO_2_.

The isolation and cultivation of endothelial progenitor cells (EPCs) was performed according to the protocol mentioned in our previous works [26,27]. After collecting peripheral blood (80 mL) from healthy donors, the peripheral blood mononuclear cells were fractionated from other blood components by centrifugation on Ficoll-Paque Plus. CD34-positive progenitor cells were obtained from the isolated peripheral blood mononuclear cells using the CD34 MicroBead kit and MACS Cell Separation System. CD34-positive EPCs were maintained and propagated in MV2 complete medium consisting of MV2 basal medium and growth supplement, supplied with 20% FBS and maintained at 37 °C in a humidified atmosphere of 5% CO_2_. The protocol were approved by the Institutional Review Board of Mackay Medical College, New Taipei City, Taiwan (reference number: P1000002), and all subjects gave informed written consent before enrollment in this study.

### 2.2. Collection of Chondrosarcoma-Conditioned Medium and Measurement of PDGF-C Production

Chondrosarcoma cells were pretreated or transfected with the pharmacological inhibitors or genetic siRNAs, then treated with visfatin. The medium was collected as conditioned medium (CM) and stored at −80 °C until use. Secreted PDGF-C was analyzed by a PDGF-C ELISA assay kit, according to the manufacturer’s procedures [28].

### 2.3. In Vitro EPC Tube Formation Assay

A 48-well plate was coated with Matrigel (BD Biosciences, Bedford, MA, USA) for the tube formation inspection. The EPCs were resuspended in the medium (50% EGM-MV2 medium and 50% chondrosarcoma cell CM) at a density of 3 × 10^4^/200 μL and added to the well. The tube formation measurement checked the discrimination and formation of capillary-like tubules on the EPC, according our previous reports [29,30].

### 2.4. Western Blot Analysis

Chondrosarcoma cells were lysed using RIPA buffer. Proteins were applied on SDS-PAGE and transferred to PVDF membranes. The membranes were then blocked with 5% non-fat milk and applied with primary antibodies (1:1000). The membranes were washed and applied with secondary antibody then visualized using the ImageQuant™ LAS 4000 biomolecular imager [31,32,33].

### 2.5. Real-Time Quantitative PCR (RT-qPCR) Analysis of mRNA and miRNA

Total RNA was isolated from cells using a TRIzol reagent. Reverse transcription was carried out using RNA (1 μg) that was reverse-transcribed into cDNA with oligo-DT primer and the Mir-X™ miRNA First-Strand Synthesis kit. qPCR was determined using SYBR Green. The primer pair sequences used were: human GAPDH, 5′-ACCACAGTCCATGCCATCAC-3′ (forward) and 5′-TCCACCACCCTGTTGCTGTA-3′ (reverse); human PDGF-C, 5′-GCCAGGTTGTCTCCTGGTTA-3′ (forward) and 5′-TGCTTGGGACACATTGACAT-3′ (reverse). qPCR assays were performed with a StepOnePlus (Applied Biosystems) [34].

### 2.6. Luciferase Reporter Assay

In order to check the 3′-UTR luciferase activity, a luciferase assay kit was used to monitor the luciferase activity. JJ012 and SW1353 were transfected with wt-PDGFC-3′-UTR or mt-PDGFC-3′-UTR luciferase plasmid (Stratagene: St. Louis, MO, USA), then activated with pharmacological inhibitors and visfatin for 24 h. Finally, luciferase activity was determined using a dual-luciferase reporter assay system (Promega, Madison, WI, USA).

### 2.7. Statistical Analysis

Statistical analyses were performed with GraphPad Prism software (San Diego, CA, USA). All values are presented as the mean ± standard deviation (SD). Significance testing of the difference between the groups was assessed by Student’s *t*-test or ANOVA followed if the *p*-value was < 0.05.

The materials and methods relating to chorioallantoic membrane (CAM) assay, matrigel plug assay, in vivo tumor xenograft model, and immunohistochemistry (IHC) staining are detailed in the Appendix A.

## 3. Results

### 3.1. Human Chondrosarcoma Tissue Is Highly Correlated with PDGF-C Levels

Evidence implicates aberrant PDGF signaling in the development and metastasis of many different cancers [35,36,37]. This study analyzed PDGF-C expression in human tissue specimens consisting of normal cartilage (*n* = 7), grade 1 chondrosarcoma (*n* = 11), grade 2 chondrosarcoma (*n* = 4), grade 3 chondrosarcoma (*n* = 6), and grade 4 chondrosarcoma (*n* = 3) (Figure 1a,b). IHC analyses revealed that levels of PDGF-C expression were significantly higher in both the low-grade (stages 1A, B) and high-grade (stages IIA, B) chondrosarcoma samples compared with the normal cartilage samples (*p* < 0.002 for all comparisons; Figure 1c,d). The tumor IHC analysis of PDGF-C expression characterized approximately one-third (29.17%) of the high-grade chondrosarcomas as having high PDGF-C expression; 8.33% exhibited low PDGF-C expression (Figure 1e).

### 3.2. Endogenous and Exogenous Visfatin Upregulates PDGF-C Production and Angiogenesis in Human Chondrosarcomas

As previous studies have determined that visfatin promotes chondrosarcoma metastasis [9], we examined whether visfatin regulates distinct angiogenic factor expression during angiogenesis in chondrosarcoma cells. Our findings revealed higher levels of *PDGF-C* gene expression than other angiogenic factors, including *ANGPD-1*, *ANGPD-2*, *ANGPD-3*, *ANGPD-4*, *PDGF-A*, *PDGF-R*, *PDGF-D*, *EGF*, *FST*, *VEGF-A*, *VEGF-C,* and *TGF* in visfatin-treated chondrosarcoma cells (Figure 2a). Therefore, PDGF-C is more important than other angiogenic factors after visfatin stimulation. We also observed dose- and time-dependent upregulation of PDGF-C mRNA synthesis (Figure 2b,c) and significant enhancements in the production of PDGF-C, according to Western blot and ELISA evaluations (Figure 2d–f). CM from JJ012 and SW1353 cells stimulated with various doses of visfatin enhanced tube formation in the EPCs, whereas anti-PDGF-C inhibited EPC tube formation (Figure 2g,h). Direct stimulation by visfatin did not affect the cell proliferation in EPCs (Appendix A). At present, the small molecule visfatin inhibitor daporinad (also known as APO-866 and FK866) has shown antiangiogenic and antitumor activity in phase II clinical trials [38,39,40]. Importantly, no reports exist as to the effect of FK866 on chondrosarcoma cell angiogenesis, in particular. We therefore investigated the effects of FK866 treatment (100 nM) on PDGF-C expression in chondrosarcoma cells. The results showed that PDGF-C mRNA and protein in JJ012 cells overexpressing visfatin were effectively and significantly downregulated by both PDGF-C shRNA knockdown and FK866 (Figure 2i,j,l,m). Similarly, EPC angiogenesis in JJ012 cells overexpressing visfatin was significantly reduced by both PDGF-C shRNA and FK866 treatment (Figure 2k; Appendix A). These results indicate that endogenous and exogenous visfatin increase levels of PDGF-C and facilitate angiogenesis in human chondrosarcoma cells.

### 3.3. PI3K/Akt Pathway Mediates Visfatin-Induced Increases of PDGF-C Production in Chondrosarcoma Cells

PI3K/Akt signaling assists with the synthesis of new blood vessels, which is important for the growth, proliferation, and survival of different tumors [41,42]. We therefore investigated whether visfatin activates PI3K/Akt signaling. In Western blot data, visfatin treatment of JJ012 cells time-dependently phosphorylated PI3K and Akt protein expression (Figure 3a). In order to investigate whether the PI3K/Akt pathway is regulated in visfatin-induced promotion of PDGF-C expression, we treated chondrosarcoma cells with Ly294002 or wortmannin (PI3K inhibitors), an Akt1/2 inhibitor, or transfected the cells with the respective siRNAs. RT-qPCR, Western blot, and ELISA evaluations revealed similar expressions (Figure 3b–f). In contrast, pretreating the cells with the PI3K inhibitors significantly inhibited visfatin-induced Akt phosphorylation (Figure 3g). These data indicate that visfatin increases PDGF-C levels in chondrosarcoma cells via PI3K/Akt signaling.

### 3.4. Visfatin-Induced Increases in PDGF-C Expression of Human Chondrosarcoma Cells Appear to Be Regulated by mTOR Signaling

As the PI3K/Akt signaling cascade is the main upstream activator of mTOR [43], which stimulates tumor angiogenesis [44], we therefore sought to determine whether visfatin mediates PI3K/Akt/mTOR signaling. The findings revealed time-dependent increases in mTOR phosphorylation in visfatin-treated JJ012 cells (Figure 4a). JJ012 and SW1353 cells were incubated with either an mTOR inhibitor or an siRNA, and found that visfatin-induced increases in PDGF-C mRNA and protein were significantly downregulated by both rapamycin and the mTOR siRNA compared with visfatin alone or control siRNA. (Figure 4b–f). PI3K and Akt1/2 inhibitors significantly abolished visfatin-induced increases in mTOR phosphorylation (Figure 4g). Thus, visfatin appears to increase PDGF-C synthesis in chondrosarcoma cells through PI3K/Akt/mTOR signaling.

### 3.5. Visfatin Increased PDGF-C Expression by Inhibiting miR-1264

Aberrant miRNA expression plays critical roles in determining the proliferation and survival of many different types of tumor cells [45,46]. Open-source software identified 25 target miRNAs as potential binding sites for candidate miRNAs in the 3′-UTR region of PDGF-C mRNA (Figure 5a). The top five low-expression candidate miRNA, including miR-1264, miR-29a-3p, miR-9-5p, miR-548-3p, and miR-944 were treated with visfatin in different concentrations (Figure 5b; Appendix A). Visfatin treatment significantly inhibited miR-1264 expression in miR-29a-3p, miR-9-5p, miR-548-3p, and miR-944 in a concentration-dependent manner (Figure 5c; Appendix A); transfecting the cells with miR-1264 mimic downregulated visfatin-facilitated increases in PDGF-C expression (Figure 5d). We then examined the relationship between miR-1264 and PI3K/Akt/mTOR signaling. Treating both cell lines with PI3K, Akt, and mTOR inhibitors, or the respective siRNAs rescued visfatin-enhanced decreases in miR-1264 expression (Figure 5e,f). We then sought to investigate whether miR-1264 mediates the 3′-UTR region of PDGF-C (Figure 5g). In an analysis of the PDGF-C 3′-UTR luciferase plasmids, visfatin enhanced wild-type, but not mutant PDGF-C 3′-UTR luciferase activity (Figure 5h). Transfecting both cell lines with a miR-1264 mimic; PI3K, Akt, and mTOR inhibitors; or siRNAs rescued visfatin-facilitated decreases in PDGF-C 3′-UTR luciferase activity (Figure 5i–k). The data suggest that miR-1264 binds to the 3′-UTR region of the human *PDGF-C* gene via PI3K/Akt/mTOR signaling and thus prevents increases in PDGF-C expression.

### 3.6. PDGF-C Knockdown and FK866 Treatment Effectively Reduced Chondrosarcoma-Associated Angiogenesis

In order to confirm that visfatin treatment increases PDGF-C expression and promotes angiogenesis in vivo, we evaluated the impact of PDGF-C knockdown and FK866 treatment upon PDGF-C expression in murine abdominal tissue. We first evaluated the effects of visfatin treatment on angiogenesis using a chorioallantoic membrane (CAM) assay. Compared with the JJ012 group, CAM angiogenesis was increased by the JJ012/visfatin group and decreased by the JJ012/visfatin/shPDGF-C and JJ012/visfatin+FK866 groups (Figure 6a–c). Next, the mouse Matrigel plug assay was used to examine the ability of visfatin to promote tumor angiogenesis in vivo (Figure 6d). The JJ012/visfatin CM plug showed high amounts of new blood vessel formation, whereas embolization plugs containing phosphate-buffered saline (PBS) mixed with Matrigel and CM from JJ012/visfatin/shPDGF-C and JJ012/visfatin+FK866 exhibited little or no vascularization (Figure 6e). Next, quantification of hemoglobin content in the mouse plug assay showed that visfatin overexpression significantly promoted chondrosarcoma angiogenesis compared with control JJ012 CM, whereas hemoglobin content was very similar for the shPDGF-C and FK866 CM conditions compared with JJ012/visfatin CM (Figure 6e). In embolized tissue sections, the numbers of angiogenesis marker CD31-positive blood vessels were significantly reduced by PDGF-C knockdown and FK866 (Figure 6f). In the xenograft mouse model (Figure 6g), JJ012/visfatin mice had higher tumor volumes at 28 days compared with control mice, while tumors were significantly suppressed by treatment with FK866 and by shPDGF-C (Figure 6h). IHC analysis revealed reductions in levels of PDGF-C protein, as well as angiogenesis markers CD31 and CD34, in the tumors from FK866-treated mice and tumors transfected with shPDGF-C (Figure 6i–l). Thus, according to the data from the CAM and Matrigel plug assays, visfatin increased PDGF-C expression and thereby stimulated angiogenesis and tumor growth.

## 4. Discussion

Chondrosarcoma is a heterogeneous malignant bone tumor with extremely high aggressiveness and the propensity to metastasize [47,48,49], with evidence supporting the process of angiogenesis as the most important factor of all [50]. Treatment of high-grade chondrosarcoma with chemotherapy or radiotherapy has a minimal therapeutic effect [51]. Previous reports have shown that high levels of visfatin expression can increase the production of NAD^+^, which stimulates the proliferation and angiogenesis of mammalian cells [52]. Our other research has identified that the adipokines resistin and leptin increase chondrosarcoma angiogenesis and metastasis via different miRNA–mRNA regulatory pathways [22,53]. Similarly, we have previously reported that visfatin facilitates the migratory and invasive properties of chondrosarcoma cells. This study has shown that visfatin enhances PDGF-C synthesis in human chondrosarcoma cells and facilitates EPC angiogenesis via PI3K/Akt/mTOR signaling. However, the main receptor of visfatin is still unknown. It has been reported that visfatin binds to CCR5 and acts as a natural antagonist of the CCR5 receptor [54]. Whether CCR5 is involved in visfatin-induced PDGF-C expression needs further examination. Our evidence suggests that visfatin blockade could serve as a small-molecule therapeutic option for chondrosarcoma.

The complex process of angiogenesis in tumor growth and metastasis is mediated by several angiogenic stimuli that regulate a balance between angiogenesis and anti-angiogenesis, with VEGF and PDGF receptors playing important roles in the coordination of angiogenesis [55,56,57]. For instance, in a case of metastatic chondrosarcoma refractory to front-line chemotherapy, second-line treatment with the PDGFR inhibitor pazopanib resulted in an objective effect lasting for more than 6 months with marked improvements in the patient’s symptomatology and common condition [58]. Subsequent clinical trials involving patients with advanced/metastatic, or surgically unresectable chondrosarcoma have shown that pazopanib has good anticancer ability, prolonging median overall survival and progression-free survival, with good tolerability [59,60]. This study therefore explored the possible benefits of PDGF-C-targeted therapy in chondrosarcoma angiogenesis. According to the study data, visfatin increased PDGF-C mRNA and protein synthesis in chondrosarcoma cell lines. Knockdown of the *PDGF-C* gene reduced visfatin-induced effects upon chondrosarcoma angiogenesis. Visfatin overexpression promoted the synthesis of PDGF-C and stimulated chondrosarcoma angiogenesis in vitro and in vivo, while treating chondrosarcoma cells with the visfatin inhibitor FK866 reduced the formation of blood vessels in chondrosarcoma tumors. We propose that visfatin promotes chondrosarcoma angiogenesis by upregulating PDGF-C expression in the tumor.

Activation of PI3K/Akt/mTOR cascades is a key factor in tumor cell development and survival, which ultimately advances angiogenesis, the development of metastasis, and therapeutic resistance [61]. These findings are supported by our study results exhibiting that visfatin stimulated PI3K, Akt, and mTOR phosphorylation, while their respective pharmacologic inhibitors inhibit visfatin-induced increases in PDGF-C expression and promotion of chondrosarcoma angiogenesis, as do the genetic siRNAs. It appears that visfatin-induced activation of PI3K, Akt, and mTOR signaling cascade controls visfatin-enhanced PDGF-C synthesis and promotes chondrosarcoma angiogenesis. We also found ERK, p38, and JNK but not FAK and PKCδ inhibitors reversed visfatin-induced PDGF-C expression. FAK and PKCδ may play visfatin-inhibiting roles. Whether FAK and PKCδ regulate upstream inhibition of miR-1264 and reduce PDGF-C expression need further examination (Appendix A).

MiRNAs function as regulators of gene synthesis, which is critical for many homeostatic processes and diseases, and for tumor development. They bind to complementary sequences in the 3′UTRs of their object mRNAs, leading to degradation or diminishment of gene translation [62]. MiR-1264 has been documented to mediate different cancer progression and motility processes [23,24,25]. In breast cancer, CircDDX21 functions as a sponge for miR-1264 and represses the EMT process [25]. In addition, SOCS2-AS1 contributes to SOCS2 expression through restraining miR-1264 in colorectal cancer progression [23]. However, no existing evidence has confirmed any regulatory activity by the PDGF-C-miRNA axis during angiogenesis in chondrosarcoma. Our study demonstrated that the miR-1264 mimic diminished visfatin-mediated PDGF-C production and EPC angiogenesis. In addition, the PI3K/Akt/mTOR signaling pathway regulated visfatin-promoted inhibition of miR-1264 synthesis. Inhibiting visfatin downregulated in vivo PDGF-C-dependent angiogenesis. We suggest that visfatin is worth targeting as a therapeutic strategy for chondrosarcoma.

## 5. Conclusions

Chondrosarcoma is associated with a very poor prognosis, with existing conventional treatments (surgery, radiation, and chemotherapy) having little benefit for survival. PDGF-C and visfatin are potent stimulators of chondrosarcoma angiogenesis and stimulate chondrosarcoma tumor aggressiveness. Our study revealed that visfatin increased PDGF-C expression and activated PI3K/Akt/mTOR signaling in chondrosarcoma cells, which subsequently downregulated levels of miR-1264 expression and promoted PDGF-C expression and angiogenesis in chondrosarcoma tumors (Figure 7). Antiangiogenic and antimetastatic therapies would therefore be of great benefit to the chondrosarcoma armamentarium. Our evidence suggests that visfatin blockade could serve as a small-molecule therapeutic option for chondrosarcoma.

## Figures and Tables

**Figure 1 cells-11-03470-f001:**
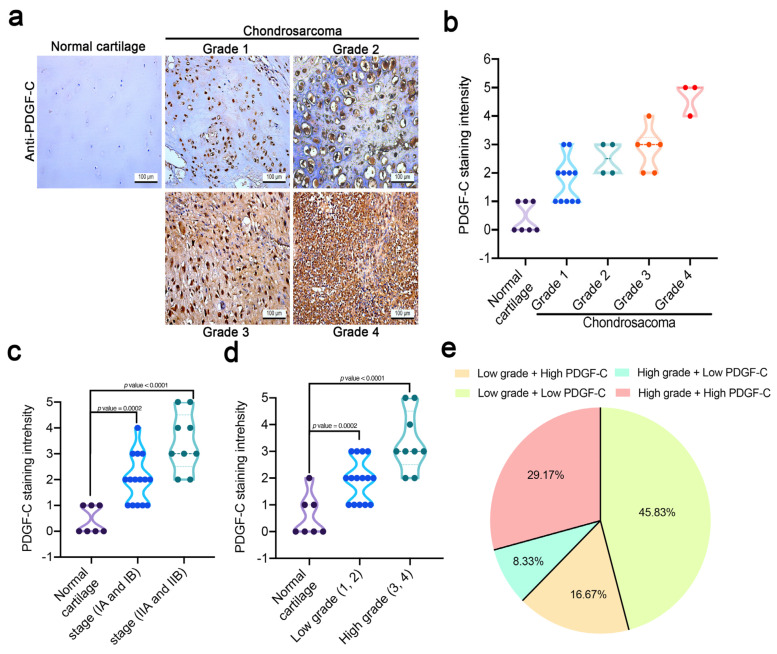
PDGF-C protein expression in normal cartilage and human chondrosarcoma specimens. (**a**) Tissue array showing PDGF-C staining intensities in normal cartilage and chondrosarcoma tissue samples; (**b**–**d**) histogram graphs showing PDGF-C staining intensities in each type of tissue specimen; (**e**) pie chart distribution of PDGF-C protein levels as determined by IHC analysis in 26 chondrosarcoma samples compared with the normal cartilage group.

**Figure 2 cells-11-03470-f002:**
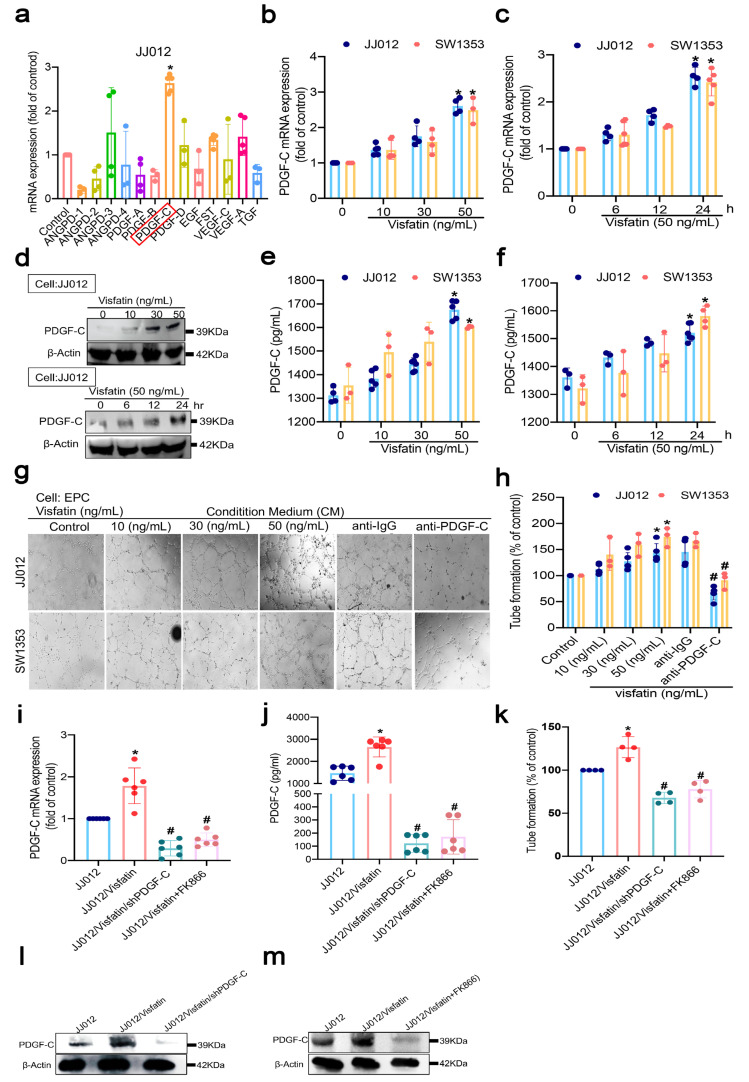
Visfatin regulated PDGF-C-dependent EPC angiogenesis. (**a**) High-dose visfatin (50 ng/mL) increased levels of angiogenesis factor expression in chondrosarcoma cells; (**b**–**f**) chondrosarcoma cells were stimulated with varying concentrations of visfatin (10–50 ng/mL) or high-dose visfatin alone (50 ng/mL) for the indicated time intervals for up to 24 h, then assayed by RT-qPCR, Western blot, and ELISA for PDGF-C expression; (**g**,**h**) chondrosarcoma cells were incubated with visfatin for 24 h and then stimulated with PDGF-C or IgG antibody (1 μg/mL) for 30 min, prior to CM collection and its application to EPCs; (**i**,**j**,**l**,**m**) protein and mRNA levels of PDGF-C expression in JJ012/visfatin/PDGF-C shRNA and JJ012/visfatin-FK866 (100 nM) cells were detected by Western blot, RT-qPCR and ELISA; (**k**) EPCs were incubated with the indicated CMs for 5 h, then photographed under a microscope to measure EPC tube formation (scale bar = 200 μm). Quantitative results are expressed as means ± SD. Compared with the control group, * *p*-values < 0.05; compared with the visfatin-treated group, ^#^ *p*-values < 0.05.

**Figure 3 cells-11-03470-f003:**
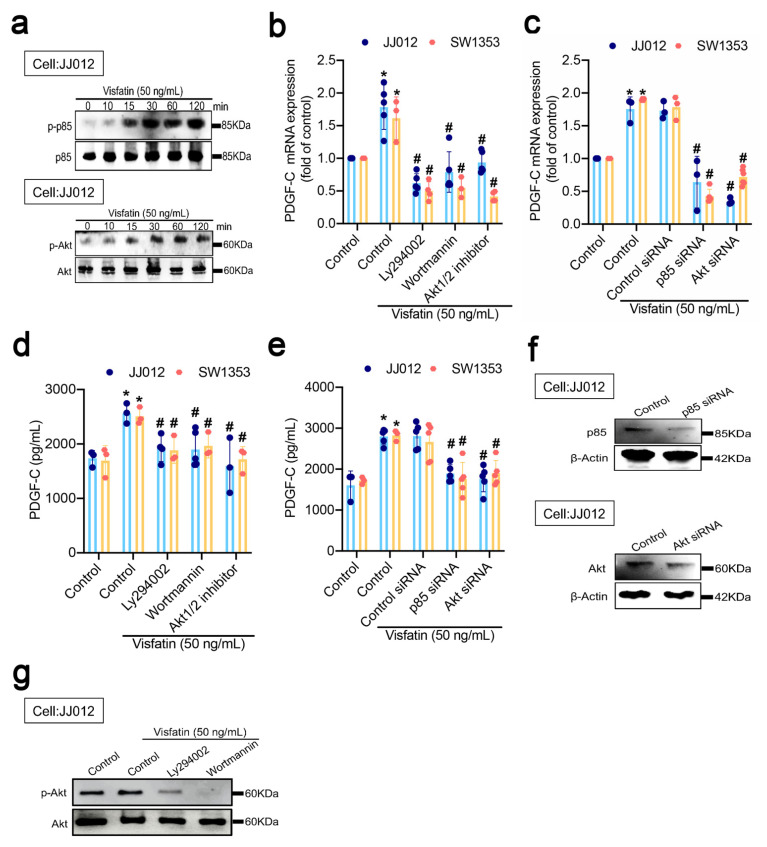
Visfatin regulated PDGF-C production in chondrosarcoma cells via the PI3K/Akt pathway. (**a**) JJ012 cells were incubated with visfatin for the indicated time intervals. PI3k and Akt phosphorylation was measured by Western blot; (**b**–**e**) cells were pretreated with Ly294002 (10 μM), wortmannin (10 μM), or an Akt1/2 inhibitor (10 μM) for 30 min, or transfected with p85 and Akt siRNAs, then stimulated with visfatin for 24 h. RT-qPCR and ELISA determined PDGF-C expression; (**f**) JJ012 cells were transfected with p85 or Akt siRNAs. Western blot detected p85 and Akt expression; (**g**) chondrosarcoma cells were pretreated with Ly294002 and wortmannin for 15 min, then stimulated with visfatin for 30 min to assess Akt phosphorylation. Quantitative results are expressed as means ± SD. Compared with the control group, * *p*-values < 0.05; compared with the visfatin-treated group, ^#^
*p*-values < 0.05.

**Figure 4 cells-11-03470-f004:**
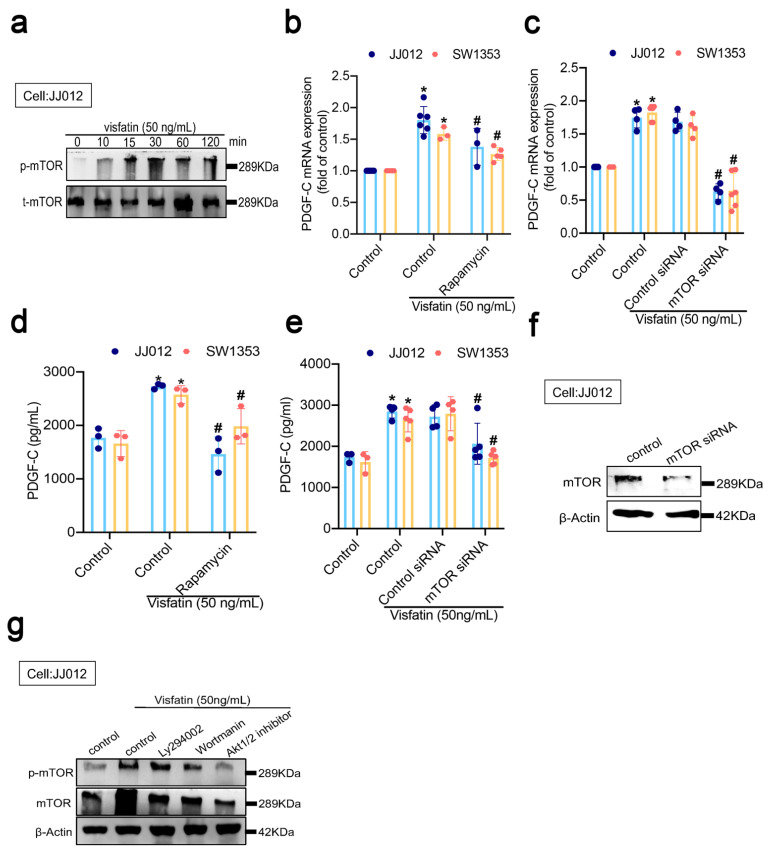
mTOR regulated visfatin-induced stimulation of PDGF-C expression in human chondrosarcomas. (**a**) JJ012 cells were incubated with visfatin for the indicated time intervals and mTOR phosphorylation was determined by Western blot; (**b**–**e**) JJ012 and SW1353 cells were pretreated with rapamycin (10 μM) or transfected with an mTOR siRNA for 30 min, then treated with visfatin for 24 h. PDGF-C expression was determined by q-PCR and ELISA; (**f**) JJ012 cells were transfected with mTOR siRNA. mTOR expression was examined by Western blot; (**g**) JJ012 cells were pretreated with Ly294002, wortmannin, or an AKT1/2 inhibitor for 15 min, then stimulated with visfatin for 30 min and mTOR phosphorylation was measured. Quantitative results are expressed as means ± SD. Compared with the control group, * *p*-values < 0.05; compared with the visfatin-treated group, ^#^ *p*-values < 0.05.

**Figure 5 cells-11-03470-f005:**
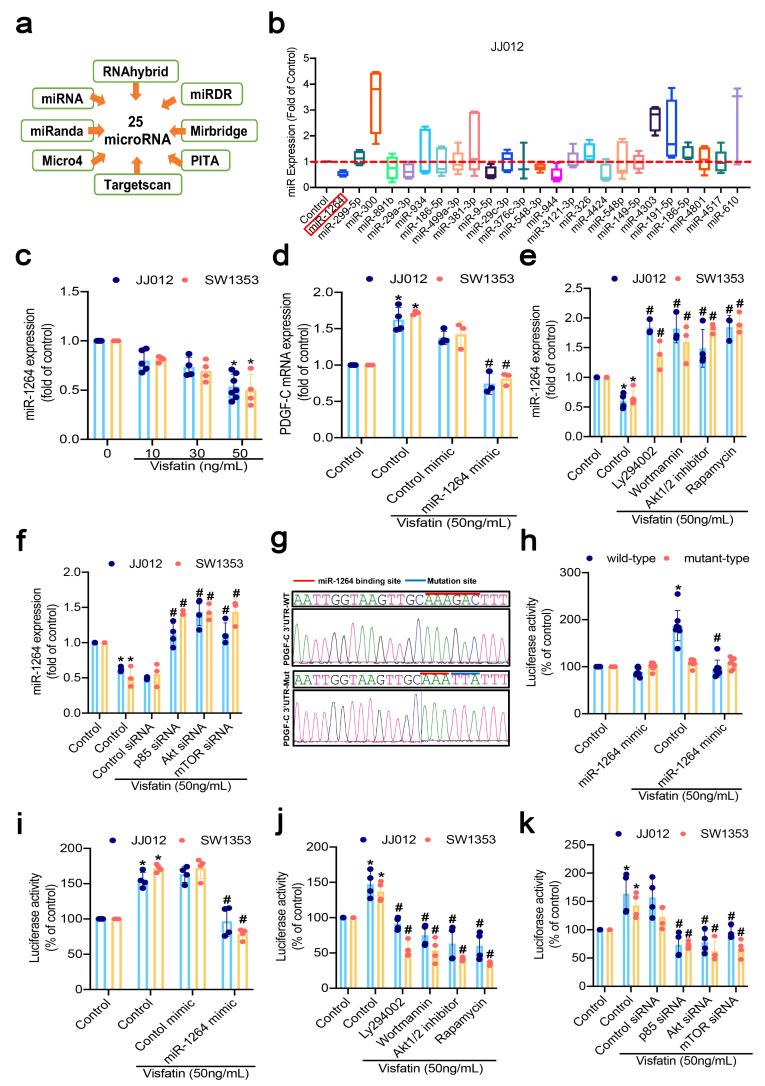
Inhibition of miR-1264 regulated visfatin-enhanced increases in PDGF-C expression of chondrosarcoma cells. (**a**) MiRNA target prediction software identified miRNAs that potentially bind to the PDGF-C 3′-UT; (**b**) chondrosarcoma cell lines were incubated with high-dose visfatin for 24 h and miRNA expression was examined by qPCR; (**c**) both cell lines were incubated with visfatin for 24 h and miR-1264 expression was examined by RT-qPCR; (**d**) The cells were transfected with miRNAs, as indicated, before being incubated with visfatin for 24 h. PDGF-C expression was determined by qPCR; (**e**,**f**) chondrosarcoma cell lines were pretreated with pharmacologic inhibitors or transfected with siRNAs, as indicated, then stimulated with visfatin for 24 h. MiR-1264 expression was analyzed; (**g**) schematic 3′-UTR representation of human PDGF-C containing the miR-1264 binding site; (**h**–**k**) PDGF-C-3′-UTR luciferase plasmids containing the miR-1264 binding site were analyzed for luciferase activity. Quantitative results are expressed as means ± SD. Compared with the control group, * *p*-values < 0.05; compared with the visfatin-treated group, ^#^ *p*-values < 0.05.

**Figure 6 cells-11-03470-f006:**
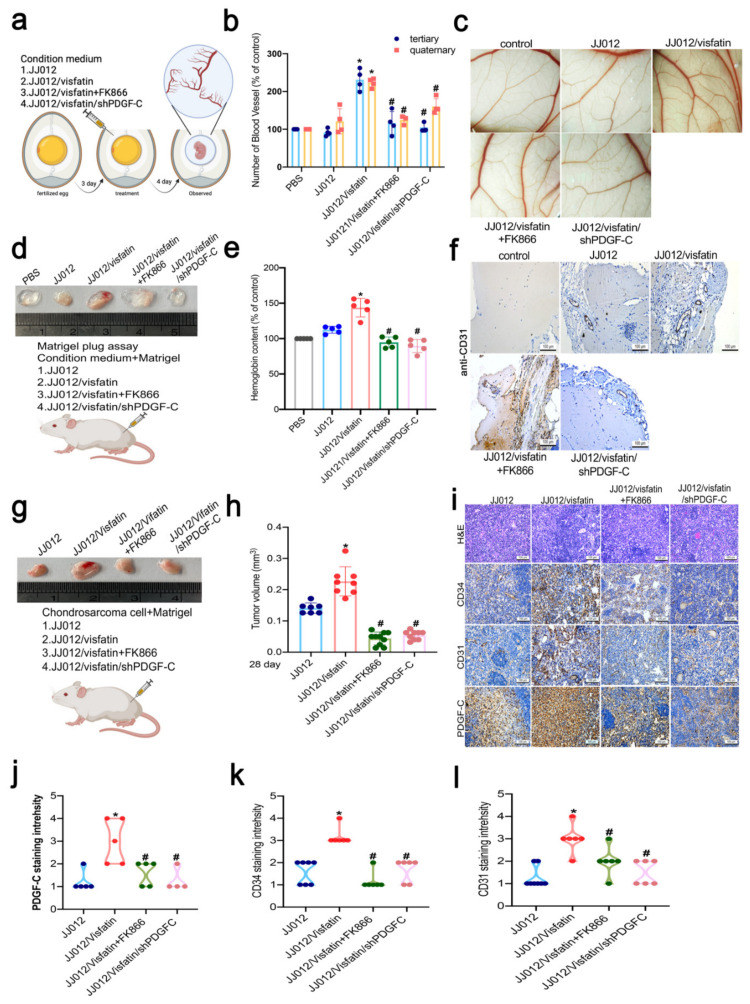
Knockdown of PDGF-C expression and FK866 treatment of human chondrosarcoma cells reduced tumor-promoted angiogenesis in vivo. (**a**–**c**) Fourteen-day-old chicken embryos were incubated with CM under specified conditions for 4 days before being excised and fixed in chorioallantoic membrane (CAM), then photographed with a stereomicroscope; (**d**–**f**) mice were injected subcutaneously with Matrigel mixed with indicated chondrosarcoma CMs for 7 days. Plugs were excised from the mice and photographed. Hemoglobin content was quantified and embedded in paraffin and the sections were immunostained with CD31 antibody; (**g**–**i**) chondrosarcoma cell lines were mixed with Matrigel, then injected into the mices’ flanks and observed for 4 weeks, before sacrificing the mice and removing the tumors. Photographs were taken of the tumors with a microscope and quantified for volume; (**j**–**l**) IHC analyses were performed of CD31, CD34, and PDGF-C expression in chondrosarcoma xenograft tissue. Quantitative results are expressed as means ± SD. Compared with the JJ012 group, * *p*-values < 0.05; compared with the JJ012/visfatin group, ^#^ *p*-values < 0.05.

**Figure 7 cells-11-03470-f007:**
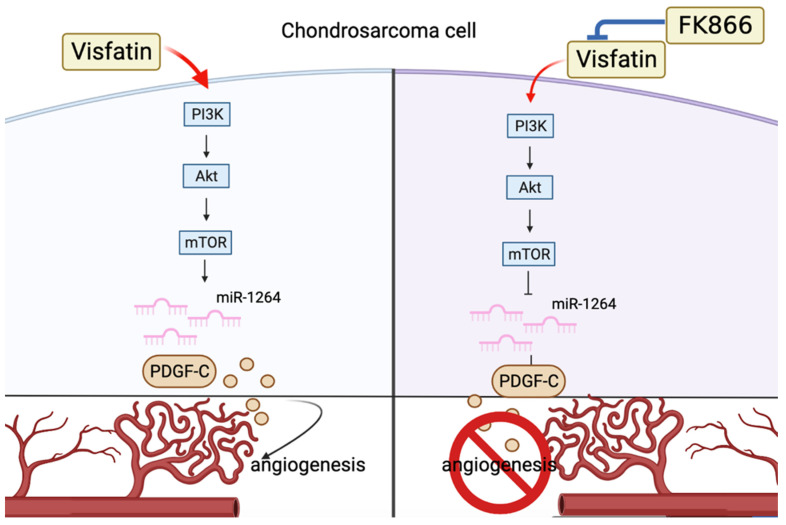
Schematic diagram of visfatin-induced increases in PDGF-C expression and EPC angiogenesis in chondrosarcoma cells. Visfatin downregulated miR-1264 via a PI3K/Akt/mTOR-dependent pathway to promote the production of PDGF-C and subsequently induce EPC angiogenesis.

## Data Availability

The raw data for this study are available from the corresponding authors.

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
