# Peer review of "Visfatin-Induced Inhibition of miR-1264 Facilitates PDGF-C Synthesis in Chondrosarcoma Cells and Enhances Endothelial Progenitor Cell Angiogenesis"

_cells, 2022, doi:10.3390/cells11213470_

Round 1

Reviewer 1 Report

The authors of this manuscript explored the importance of angiogenesis and angiogenesis-related pathways in chondrosarcoma. This is relevant and timely, since few therapeutic (novel) options exist to treat patients with this malignancy.

The manuscript needs language editing by an English native speaker or someone with command of the English language. 

Comments: 

- why have the authors simply ignored the putative importance of VEGF, and PLGF in this particular tumor setting? Are VEGF levels changed upon visfatin stimulation? There are several recent studies which implicate VEGF, and VEGFA in particular, in the pro angiogenic effects of visfatin.

- through which receptor does PDGF produce the reported effects?

- how have the authors isolated and characterized "EPCs"? This information should be provided. 

- minor point: the graphs are somewhat confusing to interpret. 

Minor point, ilustrating the English language issue: certainly in this sentence "Our evidence suggests that visfatin could serve as a small-molecule therapeutic option for chondrosarcoma", the authors mean "Our evidence suggests that visfatin blockade could serve as a small-molecule therapeutic option for chondrosarcoma".

Author Response

Eleanor Wang

Section Managing Editor,

Dear Dr. Wang,

We greatly appreciate you offering us the opportunity to revisit our manuscript, Visfatin-induced inhibition of miR-1264 facilitates PDGF-C synthesis in chondrosarcoma cells and enhances endothelial progenitor cell angiogenesis (cells-1962070), in the light of comments from your referees. We have carefully revised the manuscript according to those comments, with the use of red font to mark up all amendments in our Word file.

Reviewer 1

The manuscript needs language editing by an English native speaker or someone with command of the English language

A: The manuscript has been corrected with naïve English speaker.

Q1: - why have the authors simply ignored the putative importance of VEGF, and PLGF in this particular tumor setting?

A: The more detail information has been provided. “High levels of PDGF-C gene expression than other angiogenic factors including ANGPD-1, ANGPD-2, ANGPD-3, ANGPD-4, PDGF-A, PDGF-R, PDGF-D, EGF, FST, VEGF-A, VEGF-C and TGF in visfatin-treated chondrosarcoma cells (Fig. 2a). Therefore, PDGF-C is more important than other angiogenic factors after visfatin stimulation. (Lines 183-187)

Q2: Are VEGF levels changed upon visfatin stimulation? There are several recent studies which implicate VEGF, and VEGFA in particular, in the pro angiogenic effects of visfatin.

A: Visfatin stimulation did not significantly increase VEGF-A mRNA expression in chondrosarcoma cells (Fig. 2a).

Q3- through which receptor does PDGF produce the reported effects?

A: The information has been discussed. “However, the main receptor of visfatin is still unknown. It has reported that visfatin binds to CCR5 and acts as a natural antagonist of CCR5 receptor [54]. Whether CCR5 is involved in visfatin-induced PDGF-C expression needs further examination.” (Lines 352-355)

Q4- how have the authors isolated and characterized "EPCs"? This information should be provided. 

A: The detail information of isolated and characterized have been provided in Method. “The isolation and cultivation of endothelial progenitor cells (EPCs) was performed according to the protocol mentioned in our previous works [26,27]. After collecting peripheral blood (80 ml) from healthy donors, the peripheral blood mononuclear cells were fractionated from other blood components by centrifugation on Ficoll-Paque plus. CD34-positive progenitor cells were obtained from the isolated peripheral blood mononuclear cells using the CD34 MicroBead kit and MACS Cell Separation System. CD34-positive EPCs were maintained and propagated in MV2 complete medium consisting of MV2 basal medium and growth supplement, supplied with 20% FBS and maintained at 37°C in a humidified atmosphere of 5% CO2. The protocol were approved by the Institutional Review Board of Mackay Medical College, New Taipei City, Taiwan (reference number: P1000002), and all subjects gave informed written consent before enrollment in this study.” (Lines 102-113)

Q5- minor point: the graphs are somewhat confusing to interpret. 

A: All graphs have been reorganized.

Q6- Minor point, ilustrating the English language issue: certainly in this sentence "Our evidence suggests that visfatin could serve as a small-molecule therapeutic option for chondrosarcoma", the authors mean "Our evidence suggests that visfatin blockade could serve as a small-molecule therapeutic option for chondrosarcoma".

A: The "Our evidence suggests that visfatin could serve as a small-molecule therapeutic option for chondrosarcoma", has been corrected to "Our evidence suggests that visfatin blockade could serve as a small-molecule therapeutic option for chondrosarcoma". (Lines 410-411)

Reviewer 2 Report

In this manuscript by Song et. al., the authors shown that the visfatin downregulates levels of miR-1264 and thereby activates PI3K/Akt/mTOR signaling in chondrosarcoma cells and promotes PDGF-C expression and angiogenesis in chondrosarcoma tumors. The findings in the manuscript are well presented and identify a relevant pathway. However, there are a few concerns that the authors must address before publication.

1.     In Figure 2 (b, c), Visfatin treatment increases PDGF-C mRNA at 24h significantly, but at protein level it is not reflected (Figure 2d, lower panel)?

2.     Figure 2 (g) put clear images for SW1353 50ng/ml

3.     In Figure 2 (I, j) shPDGF-C and FK866 significantly suppressed PDGF-C mRNA and secretion. But it is not that much reflective in Figure 2 (k) EPC tube formation assay. Explanation? Also, Figure 2 (k), Images of EPC tube formation assay are missing.

4.     Figure 3 (b-e) are RT-qPCR and ELISA and no protein data. Change accordingly in figure legend.

5.     In all the experiments ELISA and RT-qPCR is done in both the cell line (JJ012 & SW1353). But why western blot is done with only one cell line

6.     Figure 5 (b), other miRNAs are also suppressed including miR-1264. But why author directly selected miR-1264 is not explained properly anywhere in the entire manuscript.

7.     Figure 6 (c) angiogenesis mainly increases tertiary and quaternary blood vessels. Why are they not shown and counted separately?

8.     Figure 6 (e) increase size of Matrigel plug assay images.

9.     Figure 6 (f) increase size of tumor xenograft images.

10.  AKT is mainly cell proliferation marker. Why authors have not checked the effect of Visfatin on EPC proliferation.

11.  In introduction and in discussion, compare and discuss your finding with previous miR-1264 data.

12.  Authors have shown that Visfatin is reducing miR-1264 expression.

Visfatin might be inhibiting upstream regulator of miR-1264 and thereby reducing its expression. Can the author perform the experiments to confirm this?

Author Response

Eleanor Wang

Section Managing Editor,

Dear Dr. Wang,

We greatly appreciate you offering us the opportunity to revisit our manuscript, Visfatin-induced inhibition of miR-1264 facilitates PDGF-C synthesis in chondrosarcoma cells and enhances endothelial progenitor cell angiogenesis (cells-1962070), in the light of comments from your referees. We have carefully revised the manuscript according to those comments, with the use of red font to mark up all amendments in our Word file.

Reviewer 2

Q1. In Figure 2 (b, c), Visfatin treatment increases PDGF-C mRNA at 24h significantly, but at protein level it is not reflected (Figure 2d, lower panel)?

A: The more consistence result has been provided in Figure 2d.

Q2. Figure 2 (g) put clear images for SW1353 50ng/ml

A: The higher clear images have been provided. (Figure 2g)

Q3.  In Figure 2 (I, j) shPDGF-C and FK866 significantly suppressed PDGF-C mRNA and secretion. But it is not that much reflective in Figure 2 (k) EPC tube formation assay. Explanation? Also, Figure 2 (k), Images of EPC tube formation assay are missing.

A: (i) The more consistence result has been provided in Figure 2k. (ii) The images of Figure 2k (EPC tube formation assay) have been provided in Supplemental data Figure S2.

Q4.  Figure 3 (b-e) are RT-qPCR and ELISA and no protein data. Change accordingly in figure legend.

A: The information has been corrected to “RT-qPCR and ELISA determined PDGF-C expression.” (Line 236)

Q5.  In all the experiments ELISA and RT-qPCR is done in both the cell line (JJ012 & SW1353). But why western blot is done with only one cell line

A: In all the experiments ELISA and RT-qPCR is done in both the cell line (JJ012 & SW1353). The ELISA assay was used to examine the secreted PDGF-C protein expression in medium. Therefore, we used the JJ012 (the most common chondrosarcoma cell line) to present the protein expression by using western blot assay.

Q6.     Figure 5 (b), other miRNAs are also suppressed including miR-1264. But why author directly selected miR-1264 is not explained properly anywhere in the entire manuscript.

A: The detail information has been added. “Open-source software identified 25 target miRNAs as potential binding sites for candidate miRNAs in the 3'-UTR region of PDGF-C mRNA (Fig. 5a). The top five low expression candidate miRNA including miR-1264, miR-29a-3p, miR-9-5p, miR-548-3p, and miR-944 were treated with visfatin different concentration (Fig. 5b; Supplemental data Figure S3). Visfatin treatment significantly inhibited miR-1264 expression than miR-29a-3p, miR-9-5p, miR-548-3p, and miR-944 in a concentration-dependent manner (Fig. 5c; Supplemental data Figure S3)” (Line 269-271)

Q7.     Figure 6 (c) angiogenesis mainly increases tertiary and quaternary blood vessels. Why are they not shown and counted separately?

A: The detail results about tertiary and quaternary blood vessels have been provided in Figure 6b.

Q8.     Figure 6 (e) increase size of Matrigel plug assay images.

A: The images have been increased sized Figure 6d.

Q9.     Figure 6 (f) increase size of tumor xenograft images.

A: The images have been increased sized Figure 6g.

Q10.  AKT is mainly cell proliferation marker. Why authors have not checked the effect of Visfatin on EPC proliferation.

A: The more detail data has been added. “Direct stimulation visfatin did not affect the cell proliferation in EPCs (Supplemental data Figure S1).” (Lines 192-193)

Q11.  In introduction and in discussion, compare and discuss your finding with previous miR-1264 data. 

A: The information has been added.

(i) In introduction: MiR-1264 has been reported to be dysregulated in colorectal cancer, laryngeal cancer, and neuroglioma [23,24]. Recently report also indicated miR-1264 regulates progression of breast cancer [25]. (Lines 82-84)

(ii) In discussion: “MiR-1264 has been documented to mediate different cancer progression and motility [23-25]. In breast cancer, CircDDX21 functions as a sponge for miR‐1264 and represses EMT process [25]. In addition, SOCS2-AS1 contributes to SOCS2 expression through re-straining miR-1264 in colorectal cancer progression [23]. However, no existing evidence has confirmed any regulatory activity by the PDGF-C-miRNA axis during angiogenesis in chondrosarcoma. Our study demonstrates miR-1264 mimic diminished visfatin-mediated PDGF-C production and EPC angiogenesis. In addition, the PI3K/Akt/mTOR signaling pathway is regulated visfatin-promoted inhibition of miR-1264 synthesis.” (Lines 391-399)

Q12.  Authors have shown that Visfatin is reducing miR-1264 expression.

Visfatin might be inhibiting upstream regulator of miR-1264 and thereby reducing its expression. Can the author perform the experiments to confirm this?

A: The more detail information has been added. “We also found ERK, p38 and JNK but not FAK and PKCδ inhibitor reversed visfat-in-induced PDGF-C expression. FAK and PKCδ may play inhibiting role of visfatin. Whether FAK and PKCδ play inhibiting upstream regulator of miR-1264 and reducing PDGF-C ex-pression are needs further examination (Supplemental data Figure S4).” (Lines 383-387)

Round 2

Reviewer 1 Report

The manuscript is acceptable in its present form, provided language (English) editing is performed. 

Reviewer 2 Report

I recommend to consider the revised manuscript for publication in the Journal "cells" 

Thank you,